# Insulin Resistance Promotes Parkinson’s Disease through Aberrant Expression of α-Synuclein, Mitochondrial Dysfunction, and Deregulation of the Polo-Like Kinase 2 Signaling

**DOI:** 10.3390/cells9030740

**Published:** 2020-03-17

**Authors:** Chien-Tai Hong, Kai-Yun Chen, Weu Wang, Jing-Yuan Chiu, Dean Wu, Tsu-Yi Chao, Chaur-Jong Hu, Kai-Yin David Chau, Oluwaseun Adebayo Bamodu

**Affiliations:** 1Department of Neurology, Taipei Medical University-Shuang Ho Hospital, New Taipei City 235, Taiwan; tingyu02139@gmail.com (D.W.); chaurjongh@tmu.edu.tw (C.-J.H.); 2Department of Neurology, College of Medicine, School of Medicine, Taipei Medical University, Taipei City 110, Taiwan; 3Taipei Neuroscience Institute, Taipei Medical University, Taipei City 110, Taiwan; 4College of Medical Science and Technology, Graduate Institute of Neural Regenerative Medicine, Taipei Medical University, Taipei City 110, Taiwan; kychen08@tmu.edu.tw (K.-Y.C.); ro8871574@gmail.com (J.-Y.C.); 5Department of Surgery, Taipei Medical University Hospital, Taipei City 110, Taiwan; wangweu@tmu.edu.tw; 6Department of Surgery, School of Medicine, College of Medicine, Taipei Medical University, Taipei City 110, Taiwan; 7Taipei Cancer Center, Taipei Medical University, Taipei City 110, Taiwan; 10575@s.tmu.edu.tw; 8Department of Hematology & Oncology, Taipei Medical University-Shuang Ho Hospital, New Taipei City 235, Taiwan; 9Department of Clinical and Movement Neuroscience, Institute of Neurology, University College London, London WC1N 3BG, UK; k.chau@ucl.ac.uk; 10Department of Medical Research & Education, Taipei Medical University-Shuang Ho Hospital, New Taipei City 235, Taiwan

**Keywords:** Parkinson’s disease, diabetes, insulin resistance, mitochondria, polo-like kinase 2, PLK2, α-synuclein, SNCA

## Abstract

**Background:** Insulin resistance (IR), considered a hallmark of diabetes at the cellular level, is implicated in pre-diabetes, results in type 2 diabetes, and negatively affects mitochondrial function. Diabetes is increasingly associated with enhanced risk of developing Parkinson’s disease (PD); however, the underlying mechanism remains unclear. This study investigated the probable culpability of IR in the pathogenesis of PD. **Methods:** Using MitoPark mice in vivo models, diabetes was induced by a high-fat diet in the in vivo models, and IR was induced by protracted pulse-stimulation with 100 nM insulin treatment of neuronal cells, in vitro to determine the molecular mechanism(s) underlying altered cellular functions in PD, including mitochondrial dysfunction and α-synuclein (SNCA) aberrant expression. **Findings:** We observed increased SNCA expression in the dopaminergic (DA) neurons of both the wild-type and diabetic MitoPark mice, coupled with enhanced degeneration of DA neurons in the diabetic MitoPark mice. Ex vivo, in differentiated human DA neurons, IR was associated with increased SNCA and reactive oxygen species (ROS) levels, as well as mitochondrial depolarization. Moreover, we demonstrated concomitant hyperactivation of polo-like kinase-2 (PLK2), and upregulated p-SNCA (Ser129) and proteinase K-resistant SNCA proteins level in IR SH-SY5Y cells, however the inhibition of PLK2 reversed IR-related increases in phosphorylated and total SNCA. Similarly, the overexpression of peroxisome proliferator-activated receptor-γ coactivator 1-alpha (PGC)-1α suppressed ROS production, repressed PLK2 hyperactivity, and resulted in downregulation of total and Ser129-phosphorylated SNCA in the IR SH-SY5Y cells. **Conclusions:** These findings demonstrate that IR-associated diabetes promotes the development and progression of PD through PLK2-mediated mitochondrial dysfunction, upregulated ROS production, and enhanced SNCA signaling, suggesting the therapeutic targetability of PLK2 and/or SNCA as potential novel disease-modifying strategies in patients with PD.

## 1. Introduction

Insulin and the insulin signaling pathways are responsible for numerous cellular functions [1]. Impaired cellular response to insulin, known as insulin resistance (IR), is a hallmark of type 2 diabetes and contributes to the systemic downstream detrimental effects of diabetes [2]. The brain is one of the major targets of insulin signaling, and neurons are particularly vulnerable to IR since impaired insulin signaling exposes neurons to enhanced metabolic stress and accelerated neuronal dysfunction [3]. Alzheimer’s disease (AD), the most common neurodegenerative disease, is increasingly being proposed as ‘type 3 diabetes’ because of the characteristic inability of brain neurons to respond to insulin in patients with AD, coupled with the accumulation of tau proteins, formation of neurofibrillary tangles (NFTs), enhanced neuro-inflammation, and neurodegeneration [4,5].

However, the role of diabetes in the pathogenesis and/or progression of Parkinson’s disease (PD) remains unclear. There are epidemiological cues that diabetes increases the risk of PD or accelerates the progression of the disease [6,7,8], however this was not substantiated in large-scale pooled meta- analysis [9]. While the negative impact of diabetes on PD continues to be suggested [10,11,12], the possible molecular basis for the probable deleterious involvement of diabetes in the neurodegeneration of dopaminergic (DA) neurons which is characteristic of PD, remains unclear.

Important pathogenetic features of dopaminergic nigral neuron degeneration in PD include the accumulation of α-synuclein (SNCA) and mitochondrial dysfunction [13]. Diabetes and cellular IR are strongly linked with systemic mitochondrial dysfunction, including in neurons [14,15]. Impaired mitochondrial biogenesis, depolarization of mitochondrial membrane potential, and increased reactive oxygen species (ROS) have been reported in neuronal cells with IR [16]. In addition, diabetic rats exhibited increased SNCA in Purkinje cells [17]. Furthermore, several anti-glycemic agents, including exenatide, a glucagon-like peptide analogue, have exerted a disease-modifying effect on PD; exenatide is presently in phase III clinical trial as a neuroprotective drug for patients with PD [18,19]. Nevertheless, the relationship between IR and the pathogeneses of PD has not been investigated directly, which makes current epidemiological findings insufficient to support the association between diabetes and PD. Against the background of piqued interests in drug-repurposing and inferable associations between diabetes, IR, and PD, there is urgent need for the discovery of efficacious novel anti-PD therapeutics drawn from the armamentarium of European Medicines Agency (EMEA)- and US Food and Drug Administration (FDA)-approved anti-glycemic drugs, as well as delineating the possible underlying molecular mechanism(s). Making use of in vitro, in vivo, and ex vivo diabetic and IR-associated PD models, the present proof-of-concept study investigated if, to what extent, and how impaired insulin signaling contributes to the development and progression of PD; determined the likelihood and nature of association between mitochondrial function and altered SNCA expression; and explored for probable therapeutic exploitability of such molecular cross-talks in patients with PD. 

## 2. Materials and Methods

### 2.1. Animal

Eight-week-old wild-type male C57BL/6 (*n* = 6, mean body weight: 24.8 ± 1.6 g), and transgenic *DAT-cre x Tfam^loxP^* MitoPark [20] (*n* = 6, mean body weight: 29.3 ± 2.5 g) mice purchased from BioLASCO (BioLASCO Taiwan Co., Ltd., Taipei, Taiwan) and the National Laboratory Animal Centre, (Taipei, Taiwan), respectively, were kept in the Taipei Medical University laboratory animal center in accordance with Taiwan regulations for experimental animals. The mice were maintained in a temperature-controlled room (~22 °C) on a 12-h light-dark cycle. The 8-week-old mice were divided into two groups, namely diabetic (*n* = 6: C57BL/6 = 3, MitoPark = 3), and non-diabetic control (*n* = 6: C57BL/6 = 3, MitoPark = 3) groups, and fed either a high-fat diet to induce diabetes [21] or continual normal diet, respectively, for up to 2 months. All animal procedures were approved and consistent with the guidelines by the Institutional Animal Care and Use Committee/Laboratory Animal Center of the Taipei Medical University (LAC-2017-0114 and LAC-2019-0077).

### 2.2. Assessment of Motor Balance and Coordination, In Vivo

Assessment of mice motor balance and coordination was performed using the beam walking and rotarod tests. For the beam walking test, diabetic and non-diabetic MitoPark mice were trained for 3 days to walk along a narrow Plexiglas beam (100 cm long, 0.5 cm wide) towards a home cage located at one end of the beam. The mean time to walk across the beam was used as a measure of motor coordination. For the rotarod test, a rotating rod (Rotarod, Ugo Basile, Washington, DC, USA) was used to evaluate both motor coordination and balance of the mice. The accelerating protocol started at a speed of 5 revolutions per minute (rpm) and reached 40 rpm within 300 s. Time to fall was the recorded primary endpoint. Both tests were carried out five times with 30 min intervals at the end of a two month high-fat diet treatment for the diabetic MitoPark mice.

### 2.3. Adipocyte-Derived Stem Cell Isolation, Culture, and Neuronal Differentiation

Visceral adipose tissue was obtained from patients with or without insulin resistant diabetes but without neurodegenerative disease during laparoscopic intra-abdominal operations or bariatric surgery. The study was approved by the Taipei Medical University Joint Institutional Review Board (N201609017).

Adipocyte-derived stem cells (ADSCs) were isolated as previously described [22]. Briefly, after washing and adipose tissue digestion by trypsin, the lysed fat cells were centrifuged, the cell pellet resuspended, and then seeded into dishes containing human mesenchymal stem cell (hMSC) SF-1 Basal medium (Cat. # MSC-SF-004) plus hMSC SF-1 Supplement (MSC-SF-005) (Biomedical Technology and Device Research Laboratories, Hsinchu City, Taiwan) to generate ADSCs. The ADSCs were sub-cultured when ~80% confluence was attained, or culture media replenished every 96 h. To induce human DA neural differentiation, ADSCs at passages 2–5 were seeded at a density of 2.5 × 10^3^ per pre-coated 35 mm cell culture dish and cultured at 37 °C in humidified 5% CO_2_ atmosphere incubator for 21–28 days; each well contained B-27™ Plus Neuronal Culture System (Cat. # A3653401, Thermo Fisher Scientific Inc., Waltham, MA, USA) supplemented with 250 ng/mL recombinant human sonic hedgehog/Shh (C24II) (Cat. # 1845-SH-100/CF, R&D Systems, Inc., Minneapolis, MN, USA), 100 ng/mL recombinant human fibroblast growth factor (FGF)-8b (Cat. # PHG0271, Thermo Fisher Scientific Inc.), 50 ng/mL recombinant basic (b)FGF (Cat. # 13256025, Thermo Fisher Scientific Inc.), and 50 ng/mL brain-derived neurotrophic factor (B-DNF; Cat. # PHC7074, Thermo Fisher Scientific Inc.). Subsequent staining of the ADSC-differentiated cells with neuron-specific class III β-tubulin (TuJ-1; Cat. # 5666S, Cell Signaling Technology, Beverly, MA, USA), a neuronal cell marker, and glial fibrillary acidic protein (GFAP; Cat. # 12389S, Cell Signaling Technology); an astrocyte marker showed that they expressed both the DA neuron phenotypic marker (~84.7%) and the astrocyte marker (15.3%) (data not provided). The Tuj-1 positive cells also exhibited strong expression of the DA neuronal markers, tyrosine hydroxylase (TH) and forkhead box protein A2 (FOXA2). This indicates that the ADSCs could differentiate into neuronal cells and astrocytes, and thus represents a proper in vitro model of neural differentiation.

### 2.4. Cell Culture

To complement our in vivo and ex vivo models and in vitro PD model, SH-SY5Y cells obtained from ATCC (American Type Cell Collection, Manassas, VA, USA) were cultured in Dulbecco’s modified Eagle’s medium (DMEM) supplemented with 10% fetal bovine serum (FBS), 1 mM pyruvate, 0.01 mM nonessential amino acids, 100 U/mL penicillin, and 100 mg/mL streptomycin in a humidified 5% CO_2_ atmosphere at 37 °C. The adherent SH-SY5Y cells were passaged or harvested by trypsinization. SH-SY5Y cells of passage numbers ≤10 were used. SH-SY5Y cells stably overexpressing peroxisome proliferator-activated receptor gamma coactivator 1-alpha (PGC-1α) were a kind gift from Dr. K.Y. Chau (Institute of Neurology, University College London, London, UK) [23]. PGC-1α-overexpressing cells were maintained under the same conditions as wild-type SH-SY5Y cells with additional Geneticin selective antibiotic (G418; Cat. # 10131035, Thermo Fisher Scientific Inc.) for selection of stably transfected clones. For induction of cellular insulin resistance, after washing with 1X PBS, the cells were incubated in fresh high glucose DMEM (4.5 g/L) containing 0.5% bovine serum albumin (BSA), and 100 nM insulin (Cat. # 109-50MG, Sigma-Aldrich, Inc., St. Louis, MO, USA) for 24 h, as previously described [24]. 

### 2.5. Fluorescence-Activated Cell Sorting Flow Cytometry

For molecular characterization, ADSCs were stained with a PerCP-Cy-conjugated anti-CD73 (Cat. # 561260, BD Pharmingen), APC-conjugated anti-CD90/Thy1 (Cat. # 559869, BD Pharmingen), PE-conjugated anti-CD11b (Cat. # 557321, BD Pharmingen), and PE-conjugated anti-CD19 (Cat. # 561741, BD Pharmingen) antibodies purchased from BD Bioscience (BD Biosciences, San Jose, CA, USA) and diluted at 1:125. Data were analyzed with the BD FACSDiva software v. 8.0.1 using a BD LSRFortessa cell analyzer (BD Biosciences, San Jose, CA, USA). 

### 2.6. Western Blot Analysis

After IR SH-SY5Y and non-IR SH-SY5Y cells were collected by centrifugation at 12,000 rpm for 5 min and washed twice with cold 1X PBS, the cell pellets were re-suspended in 1% Triton X-100/Halt™ protease inhibitor cocktail (100X) (Cat. # 78429, Thermo Fisher Scientific Inc.) containing lysis buffer, and equal amounts of protein were loaded per well for electrophoresis using NuPAGE- sodium dodecyl sulfate polyacrylamide gel electrophoresis (SDS-PAGE) under reducing conditions. Thereafter, the proteins were transferred to Hybond-enhanced chemiluminescence (ECL) nitrocellulose membranes, blocked in 5% skim milk in Tris-buffered saline with 0.1% Tween 20 (TBST), and then incubated with primary antibodies against anti-α-synuclein (#2642, 1:1000, Cell Signaling Technology), anti-Ser129 phosphorylated α-synuclein antibody (ab51253, 1:1000, Abcam plc., Burlingame, CA, USA), anti-polo-like kinase 2 (PLK2) (ab176392, 1:1000, Abcam plc.), and anti-β-actin antibody (MAB1501, 1:10,000, MilliporeSigma, Burlington, MA, USA) overnight at 4 °C. The blots were then washed with TBST, incubated with appropriate horseradish peroxidase-conjugated secondary antibody for 1 h, and washed again with TBST, before blot visualization using enhanced chemiluminescence (ECL) reagents and the BioSpectrum Imaging System (UVP, Upland, CA, USA). 

### 2.7. Protease K Digestion of α-Synuclein

To assess whether proteinase K-resistant α-synuclein increased under the condition of IR, pretreated SH-SY5Y cells were harvested, collected as a cell pellet, and treated with 0.2 μg/mL of proteinase K (Cat. # 124568, Sigma-Aldrich, Inc.) for 10 min at 37 °C. Following treatment, proteinase K was neutralized by egtazic acid (EGTA; Cat. # E8145-50G, HPLC ≥97%, Sigma-Aldrich, Inc.) and proteinase inhibitors. The samples were then subjected to analysis by Western blot for α-synuclein and β-actin.

### 2.8. Oxidative Stress Assessment

In PD cells, oxidative stress was measured using fluorogenic dye 2′,7′-dichlorodihydrofluorescein diacetate (H_2_-DCFDA; Cat. # D6883, HPLC ≥97%, Sigma-Aldrich, Inc.), which is oxidized by intracellular ROS. Treated SH-SY5Y cells were incubated with 50 μM H_2_-DCFDA for 30 min and then washed with 1X phosphate buffered saline (PBS; Cat. # P7059, Sigma-Aldrich, Inc.), and the fluorescent compound was detected in a fluorescence microplate reader with excitation and emission wavelengths of 495 and 529 nm, respectively. For differentiated human DA (d-hDA) neurons, oxidative stress was detected by live imaging using CellROX deep red reagent (Cat. # C10422, Thermo Fisher Scientific Inc.). Treated d-hDA neurons were applied with 5 μM CellROX for 30 min and then washed with 1X PBS twice. Afterwards, the fluorescent intensity was assessed through laser confocal microscopy.

### 2.9. Mitochondrial Membrane Potential (ΔΨm) Assessment

SH-SY5Y cells were harvested and washed with cold PBS. Re-suspended cells were incubated (10 min, 37 °C) in culture medium containing 5,5′,6,6′-tetrachloro-1,1′,3,3′-tetraethylbenzimidazol-carbocyanine iodide (JC-1; Cat. # T3168, Thermo Fisher Scientific Inc.) at a final concentration of 5 μg/mL. JC-1 is a lipophilic fluorescent dye that accumulates in the mitochondrial membrane, where it forms aggregates dependent on the ΔΨm. In intact living cells, JC-1 aggregates are profoundly fluorescent orange in mitochondria, which decrease with ΔΨm collapse. After staining, cells were washed again and then the fluorescent compound was detected in a fluorescence microplate reader at excitation and emission wavelengths of 535 and 585 nm, respectively (for aggregate form) or 485 and 535 nm, respectively (for monomer form). For d-hDA neurons, cells were incubated with JC-1 (5 μg/mL, 30 min), washed with PBS twice, and then the fluorescent intensity was assessed under a laser confocal microscope.

### 2.10. Immunohistochemistry

Single-labeling immunohistochemistry was performed using the Dako REAL™ EnVision™ Detection System, Peroxidase/DAB+ system (Cat. # K500711-2, Agilent Technologies, Santa Clara, CA, USA) as previously described [25]. Briefly, 14 μm sections of the mesencephalic regions largely containing the substantia nigra distributed on glass slides after being serially sectioned using a cryostat, were stained with anti-tyrosine hydroxylase antibody (Cat. # MAB318, 1:200–400, Millipore, Burlington, MA, USA) and anti-α-synuclein antibody (Cat. # 2642, 1:1000, Cell Signaling Technology).

### 2.11. Immunocytochemistry (IHC)

Immunocytochemistry was performed as previously described [26]. Briefly, after fixing cells with 4% paraformaldehyde in phosphate buffered solution (PBS) (Cat. # AAJ19943K2, Thermo Fisher Scientific Inc.), they were permeabilized by adding cold methanol and then probed with the indicated primary antibodies. Antibodies used are as follows: anti-α-synuclein (#2642, 1:1000, Cell Signaling Technology), anti-phospho-α-synuclein (Ser129) (ab51253, 1:1000, Abcam plc.), anti-tyrosine hydroxylase (MAB318-AF488, 1:100, Millipore), anti-FOXA2 (ab108422, 1:300, Abcam plc), anti-phospho-IRS1 (Ser302) (Cat. # 2384, 1:1000, Cell Signaling Technology), and anti-IRS-1 (Cat. # 2382, 1:1000, Cell Signaling Technology). Fluorescence intensities from images of randomly selected microscopic fields of cells were semi-quantitatively analyzed based on staining positivity indices estimation using the NIH ImageJ software (https://imagej.nih.gov/ij/) as previously described by Jensen [27].

### 2.12. RNA Extraction, cDNA Synthesis, and Quantitative Real-Time Polymerase Chain Reaction

Total RNA was isolated using the PureLink RNA Mini Kit (Cat. # 12183018A, Thermo Fisher Scientific Inc.). Total RNA concentration and purity were determined using the NanoDrop™ ND1000 spectrophotometer (Nyxor Biotech, Paris, France). Equal amounts (2 μg) of total RNA were treated with reverse transcriptase to obtain cDNA using the SuperScript VILO cDNA Synthesis Kit (Cat. # 11754050, Thermo Fisher Scientific Inc.). All procedures were performed following the manufacturers’ instructions. To quantify the mRNA expression levels, equal amounts (0.5 μg) of cDNA were mixed with TaqMan Fast Advanced Master Mix (Cat. # 4444556, Thermo Fisher Scientific Inc.) and primers were used as follows: PLK2 (forward: 5′-GCTGATGTCTGGCTGTTCATCAG-3′; reverse: 5′-CTTCCCTGTAGATCTCACAGTG-3′); and GAPDH (forward: 5′-ACCCAGAAGACTGTGGATGG-3′; reverse: 5′-TTCAGCTCAGGGATGACCTT-3′). Glyceraldehyde 3-phosphate dehydrogenase (GAPDH) was amplified as an internal control. qPCR was conducted at 95 °C for 10 min, followed by 40 cycles at 95 °C for 15 s and at 60 °C for 1 min. The threshold-crossing value was noted for each transcript and normalized to the internal control. The relative quantization of each mRNA sample was performed using the comparative C_T_ method. For fold inductions, we used the formula 2^−(ΔΔCt)^, where ΔΔCt is the ΔCt_(PLK2)_ − ΔCt_(GAPDH)_ and C_T_ is the cycle at which the threshold is crossed. Experiments were performed three times independently in quadruplicate using the CFX96 Touch Real-Time PCR Detection System (Bio-Rad Laboratories, Inc., Hercules, CA, USA).

### 2.13. Polo-Like Kinase 2 Activity Assays

Polo-like kinase 2 (PLK2) activity was assessed using the CycLex Polo-like kinase 2 Assay/Inhibitor Screening Kit (CycLex Co., Nagano, Japan) following the manufacturer’s instructions. In brief, cell lysates were loaded onto plates precoated with PLK2 substrate and then threonine residue phosphorylation was detected with anti-phospho-threonine polyclonal antibodies.

### 2.14. Confocal Imaging

The cultured d-hDA neurons plated on the precoated glass dish were imaged using a Leica TCS SP5 confocal microscope (Leica Microsystems, Mannheim, Germany) with a 63 × 1.25 1.4-numerical aperture oil immersion objective, and images were processed using ImageJ measurement tools. For live cell imaging, we used CellROX green reagent (Cat. # C10444, Thermo Fisher Scientific Inc.) to detect cytosolic ROS production, and JC-1 to evaluate ΔΨm.

### 2.15. Deconvolution Microscopy Imaging

For deconvolution microscopy, after immunofluorescent staining of midbrain sections of the mice, images were obtained using a DeltaVision deconvolution microscope (GE Healthcare, Phoenix, AZ, USA) equipped with 60×/1.42-N.A. oil immersion objective lens. Stacks of optical section images were collected for all fluorochromes, deconvoluted using SoftWorX software v.7.0.0, and analyzed using the ImageJ software.

### 2.16. Datasets and Bioinformatics Analyses

Transcript expressions of PD-pathognomonic markers were retrieved from the Kyoto Encyclopedia of Genes and Genomes (KEGG: https://www.genome.jp/kegg-bin/get_htext#C222). Gene Expression Omnibus (GEO) genomic and associated clinicopathological datasets were assessed from the European Molecular Biology Laboratory–European Bioinformatics Institute (EMBL-EBI) ArrayExpress platform (https://www.ebi.ac.uk/arrayexpress/experiments/). Datasets used in the study were E-GEOD-20291 (https://www.ebi.ac.uk/arrayexpress/experiments/E-GEOD-20291/), E-GEOD-20292 (https://www.ebi.ac.uk/arrayexpress/experiments/E-GEOD-20292/), E-GEOD-13070 (https://www.ebi.ac.uk/arrayexpress/experiments/E-GEOD-13070/), and E-GEOD-20168 (https://www.ebi.ac.uk/arrayexpress/experiments/E-GEOD-20168/). The heatmaps of genomic data were generated using the Euclidean distance function and average hierarchical linkage clustering method parameters on the Heatmapper (http://heatmapper.ca/). 

The Search Tool for the Retrieval of Interacting Genes/Proteins (STRING) version 11.0 online platform (https://string-db.org) was used for protein–protein network visualization, by initial manual inputting of proteins of interest in the ‘multiple proteins by names’ with subsequent auto-expansion of network protein components and ‘Homo’ organisms. “Evidence” defined the network edges, “Text-mining, Experiments, and Databases” defined the active interaction sources, and the minimum required interaction score was set at “highest confidence (0.900)”. 

Bioinformatics-aided molecular docking was performed using the Schrödinger PyMOL 2.3 (https://pymol.org/2/). Three-dimensional (3D) molecular structures of SNCA (PDB: 6A6B) and PLK2 (PDB: 4RS6) in PDB format were obtained from the Research Collaboratory for Structural Bioinformatics Protein Data Bank (https://www.rcsb.org/), while the 3D structure of reactive oxygen species modulator 1 (ROMO1) was generated using the (PS)^2^: protein structure prediction server version 3.0 (http://ps2v3.life.nctu.edu.tw/).

### 2.17. Statistical Analysis

All data are presented as mean ± standard error of mean (SEM) of assays performed at least three times in triplicate. Comparison between two groups was performed using two-tailed Student’s *t* tests and one-way analysis of variance (ANOVA) was used to compare ≥3 groups. All statistical analyses were performed utilizing the GraphPad Prism version 7.00 for Windows (GraphPad Software Inc., La Jolla, CA, USA). A *p*-value of <0.05 was considered to be statistically significant.

## 3. Results

### 3.1. Patients’ Genomic Landscape Indicates a Nosological Association between Diabetes and PD, In Silico

Against the background of suggested links between insulin signaling and neurodegenerative diseases [3,4,5,6,7,8,10,11,12], as well as the pathological relevance of the putamen, prefrontal cortex, and substantia nigra in PD [28], we probed for probable association between diabetes-related insulin resistance and PD using bioinformatics approaches. Results of our re-analyses of genomic data from Homo sapiens, E-GEOD-20291 PD cohort (*n* = 30 samples, 22,283 genes) (https://www.ebi.ac.uk/arrayexpress/experiments/E-GEOD-20291/) we observed that the transcript expression of KEGG-curated markers of PD (https://www.genome.jp/kegg-bin/get_htext#C222), including SNCA, parkin (PRKN), ubiquitin C-terminal hydrolase L1 (UCHL1), PTEN-induced kinase 1 (PINK1), Parkinson-associated deglycase (PARK7/DJ1), leucine-rich repeat kinase (LRRK)2, coiled-coil-helix-coiled-coil-helix domain containing 2 (CHCHD2), and high temperature requirement protein A2 (HTRA2), were co-enhanced with biomarkers of IR as documented by Park et al. [29], such as insulin (INS), INS receptor (INSR), polo-like kinase 2 (PLK2), tyrosine hydroxylase (TH), forkhead box protein A2 (FOXA2), insulin receptor substrate (IRS1), retinol-binding protein (RBP)4, peroxisome proliferator-activated receptor gamma coactivator 1-alpha (PPARGC1A/PGC-1α), alpha-2-HS-glycoprotein (AHSG), fatty acid binding protein (FABP)4, retinoic acid receptor responder (RARRES)2, fibroblast growth factor (FGF)21, and fibronectin type III domain containing 5 (FNDC5) mRNAs, but inversely correlated with adiponectin (ADIPOQ) and PPARGC1A in the putamen (Figure 1A), prefrontal cortex area 9 (Figure 1B), and whole substantia nigra (Figure 1C) of patients with PD, compared to their non-PD counterparts. Similarly, re-analysis of the E-GEOD-13070 clinical cohort on human IR and thiazolidinedione (TZD)-mediated insulin sensitization (IS) (*n* = 11 samples, 54,675 genes) (https://www.ebi.ac.uk/arrayexpress/experiments/E-GEOD-13070/) demonstrated enhanced co-expression of IR and PD biomarkers in IR subjects, converse to their low expression in the IS group (Appendix A). The heatmaps of genomic data were generated using the Euclidean distance function and average hierarchical linkage clustering method parameters on the Heatmapper (http://heatmapper.ca/). Notably, using the ‘multiple proteins by names’ and ‘Homo’ organism functions of the STRING ver.11.0 (https://string-db.org) protein–protein network visualization platform, we demonstrated molecular interaction between the molecular markers of PD including SNCA, TH, PLK2, and IR markers such as FOXA2, IRS1, INS, INSR, PPARGC1A/PGC-1α, and ADIPOQ, with an average node degree of 9.8°, average local clustering coefficient of 73.8%, and protein–protein interaction (PPI) enrichment *p*-value <1 × 10^−16^ (Figure 1D); where ‘nodes’ are ‘proteins’ and ‘edges’ are ‘interactions’. Computation of 147 edges, where only 18 were expected, indicate that the PD-IR proteins exhibit more molecular interactions among themselves than would be expected for a random set of proteins of similar size, drawn from the genome; this enrichment pattern indicates that the PD and IR proteins are at least, in part, biologically connected as a biopathological entity. These data, at least in part, indicate that diabetes-related INS, by interacting with and/or regulating SNCA, PLK2, TH, INSR, IPS1, PPARGC1A, and ADIPOQ, drives IR and modulates PD-pathognomonic neurodegeneration.

### 3.2. Diabetes Contributes to the Progression of PD by Amplifying the Loss of TH^+^ Dopaminergic Neurons and Enhancing SNCA Expression, In Vivo

To determine the role of diabetes in the pathogenesis of PD, using murine diabetic models, we probed for the possible presence of PD-specific genomic and phenotypic traits in diabetic mice. As already alluded, diabetes was induced by feeding C57BL/6 mice a high-fat diet for two months; an approach which enhances the loss of DA neurons in toxin-induced Parkinsonism, in vivo [30]. We observed that the induction of diabetes (Appendix A) did not result in any appreciable loss of dopaminergic (DA) neurons in the midbrain substantia nigra of the diabetic C57BL/6 mice (Figure 2A), however, there was an increase in the immunoreactivity of SNCA in the substantia nigra of the diabetic MitoPark mice (Figure 2B). Consistent with this, immunofluorescence staining results showed reduction in the pool of TH-positive neurons (Appendix A), with concomitant enhanced expression of SNCA in both TH-positive and -negative neurons of diabetic MitoPark mice, compared to their non-diabetic control counterparts (~38%, *p* < 0.01) (Figure 2C).

Understanding that despite the association of high-fat diet-induced diabetes with significant loss of DA neurons in toxin-induced in vivo PD models [30,31], these toxin-induced PD models cannot recapitulate the slow, progressive degeneration of DA neurons; therefore, we also used the transgenic MitoPark mice PD models which harbor homozygous disruption of the mitochondrial transcription factor A (Tfam) in the DA neurons and recapitulate the mitochondrial dysfunction in adult-onset sporadic PD, to determine the role of diabetes in PD. Against the background that MitoPark mice usually do not exhibit high SNCA expression [20], we demonstrated that the induction of diabetes enhanced the expression of SNCA protein in the midbrain tectum of the diabetic MitoPark mice compared to the control non-diabetic MitoPark mice (27.4%, *p* < 0.05) (Figure 2D). Furthermore, we demonstrated that induction of diabetes significantly increased the expression of SNCA in the substantia nigra of the diabetic MitoPark mice, compared with the control MitoPark mice (39.2%, *p* < 0.05) (Figure 2E). We also observed that the SNCA^high^ diabetic MitoPark mice took a longer time to walk across the narrow beam (22%, *p* < 0.05) (Figure 2F), and stayed a shorter length of time on the rotarod (37%, *p* < 0.05), compared to their control counterparts (Figure 2G). These data indicate, at least in part, that diabetes contributes to the course of PD and associated loss of motor coordination through induced loss of TH^+^ DA cells and enhanced expression of SNCA, which are relevant to motor deficit in PD.

### 3.3. Insulin Resistance Is Associated with Increased Expression of SNCA and Mitochondrial Dysfunction in Human Differentiated Dopaminergic Neurons

For functional and phenotypic characterization of the fibroblast-like ADSCs (Figure 3A; also see Appendix A) generated from patients’ adipose tissue as earlier described [22], on day 21 of neuronal differentiation of the ADSCs, we observed that the d-hDA neuronal cells had neurite outgrowth (Figure 3B) which exhibited strong expression of the DA neuronal markers, TH and FOXA2 (Figure 3C). Similar to the in vivo models, in the d-hDA neurons, the induction of IR resulted in increased SNCA immunoreactivity, compared to the non-IR cells (18.4%, *p* < 0.05) (Figure 3D,E). Moreover, since mitochondrial dysfunction is pathognomonic of PD, we evaluated mitochondrial function in the d-hDA neurons with IR using live laser confocal imaging. We demonstrated an elevated ROS level (46.6%, *p* < 0.001) (Figure 3F,G) and JC-1 fluorescence-based depolarization of ΔΨm (22.2%, *p* < 0.001) (Figure 3H,I) in the IR d-hDA neurons, compared to the non-IR cells. These data do indicate that IR induces the elevated expression of SNCA and mitochondrial dysfunction in d-hDA neurons, a bio-duo that is mechanistically intrinsic to the development and progression of PD.

### 3.4. Peroxisome Proliferator-Activated Receptor Gamma Coactivator 1-Alpha (PPARGC1A/PGC-1α) Impedes IR-Induced Mitochondrial Dysfunction

Mechanistically, since increased p85α expression and IRS1 serine phosphorylation are essential for the induction of clinically-apparent IR [32], we demonstrated significant increase in the p-IRS1/IRS1 ratio (123%, *p* < 0.05) in IR SH-SY5Y cells (Figure 4A). In parallel assays we also demonstrated elevated levels of mitochondrial dysfunction-associated DCFDA oxidation in the IR cells, compared to the non-IR control cells (52.1%, *p* < 0.01) (Figure 4B), which are suggestive of increased ROS, as well as enhanced mitochondrial depolarization evidenced by a 11.1% (*p* < 0.01) decrease in the JC-1 aggregate (red fluorescence)/monomer (green fluorescence) ratio in the IR cells, compared to their non-IR counterparts (Figure 4C). Furthermore, because of the documented critical role of PGC-1α/PPARGC1A in the modulation of insulin signaling and as a master regulator of mitochondrial biogenesis in the brain [33], we investigated the effect of IR on PGC-1α. Our results showed that induction of IR elicited significant downregulation of PGC-1α protein level in the neuronal cells (26.6%, *p* < 0.05) (Figure 4D,E). In addition, we observed that while IR wild-type cells exhibited an ~100% increase in DCFDA oxidation levels compared to control SH-SY5Y cells (*p* < 0.01), no apparent difference was noted between control and IR PGC-1α-overexpressed SH-SY5Y cells (Figure 4F); this indicates, at least in part, that the ectopic expression of PGC-1α protects against IR-elicited, enhanced oxidized DCFDA-related oxidative stress and mitochondrial dysfunction.

### 3.5. IR Elicits Increased Total, Ser129-Phosphorylated, and Proteinase K-Resistant SNCA through a Polo-Like Kinase 2-Dependent Mechanism

We demonstrated that the induction of IR resulted in increased expression of total SNCA (32.1%, *p* < 0.05) and p-SNCA (Ser129) (114%, *p* < 0.01) in SH-SY5Y cells (Figure 5A,B), consistent with evidence that Ser129 phosphorylation is involved in the pathologic oligomerization of SNCA, making SNCA more resistant to digestion by protease K [33]. We also showed that compared with the non-IR control SH-SY5Y cells, the IR cells exhibited upregulated SNCA protein expression levels in the presence (1.74-fold, *p* < 0.01) or absence (1.98-fold, *p* < 0.01) of proteinase K with a statistically non-significant 1.14-fold difference in SNCA expression between the IR PK(−) and PK(+) cells (Figure 5C,D), suggesting that the induction of IR reduces the sensitivity of SNCA to the protein-degrading effect of proteinase K. Moreover, concordant with documented implication of PLK2 in the phosphorylation of SNCA at Ser129 [34,35], we demonstrated that IR elicits a significant increase in PLK2 kinase activity (38.9%, *p* < 0.001), and this is amply reversed by treatment with 0.0625 nM BI-2536, a potent inhibitor of PLK2 (Figure 5E). Similarly, exposure to BI-2536 suppressed the expression of p-SNCA (Ser129) in both wild-type and IR SH-SY5Y cells (Figure 5F). These data indicate, at least in part, that PD-pathognomonic increase in total, Ser129-phosphorylated, and proteinase K-resistant SNCA is mediated by enhanced expression of PLK2.

### 3.6. PGC-1α Overexpression Represses IR-Enhanced PLK2 Activity and Subsequent Increase in SNCA

In parallel assays, we demonstrated that IR-induced increase in PLK2 activity (shown in Figure 4E) is concomitant with upregulated expression of PLK2 at the protein (52.4%, *p* < 0.01) (Figure 6A,B), and mRNA (38.9%, *p* < 0.01) (Figure 6C) levels in the IR SH-SY5Y cells compared to their non-IR control counterparts. Furthermore, because of the critical role of PGC-1α in the interplay between insulin signaling, oxidative stress, and mitochondrial dysfunction [33], and the implication of PLK2 mRNA in the induction of oxidative stress [36], we examined the effect of altered PGC-1α expression on PLK2. Our results indicated that PGC-1α overexpression repressed IR-induced increase in PLK2 activity (Figure 6D). In addition, PGC-1α overexpression elicited the downregulation of total and Ser129-phosphorylated SNCA levels in IR SH-SY5Y cells (Figure 6E). These data indicate that IR-enhanced PLK2 activity and subsequent increase in SNCA is repressed by the overexpression of PGC-1α.

### 3.7. IR-Driven SNCA-PLK2-ROS Signaling Is Implicated in PD Development and Progression

Having demonstrated that the induction of IR is associated with PD-specific mitochondrial oxidative stress and increased expression of SNCA and PLK2 expression, concordant with earlier results, using bioinformatics-aided molecular docking (Schrödinger PyMOL 2.3 (https://pymol.org/2/)), we demonstrated that SNCA (PDB: 6A6B) interacts and binds with PLK2 (PDB: 4RS6) directly with a shape complementarity/docking score of –268.90, and ligand root-mean-square deviation (RMSD) of 47.26Å (Figure 7A). Consistent with the complicity of mitochondrial dysfunction and oxidative stress in PD, we probed and computationally demonstrated that upon formation of the SNCA/PLK2 complex, the activated SNCA actively interacts with and binds to reactive oxygen species modulator 1 (ROMO1) with a docking score of –244.41, and ligand RMSD of 59.80Å (Figure 7B). ROMO1 3D structure was generated using the (PS)^2^: protein structure prediction server version 3.0 (http://ps2v3.life.nctu.edu.tw/). Put together, our data, as illustrated in our schematic abstract, indicate an association between IR and PD, with IR promoting PD through aberrant expression of SNCA, enhanced mitochondrial oxidative stress, mitochondrial dysfunction, and deregulation of the PLK2 signaling (Figure 7C). 

## 4. Discussion

The present proof-of-concept study provides some insight into the role of diabetes-related insulin resistance in the development and progression of PD using in vitro, ex vivo, and in vivo models. Our results show that diabetes-related IR consistently induces or enhances bio-phenomena that are pathognomonic of PD, namely aberrant SNCA expression, oxidative stress, and mitochondrial dysfunction. These data are consistent with epidemiological suggestions that diabetes may be associated with an increased risk of PD. Diabetes is associated with impaired insulin signaling in the brain [37]. Chronic hyperinsulinemia induces IRS1 hyperactivity in neurons and eventually leads to IR with its associated pathogenic effects, including mitochondrial dysfunction, oxidative stress, abnormal glucose metabolism, inflammation, protein aggregation, and defective neurogenesis [37]. These effects are associated with neurodegeneration, which underlies the evolution of PD. Diabetes is increasingly implicated in many ageing and neurodegenerative diseases [6,7,10,38].

As illustrated in our schematic abstract, for the first time and to the best of our understanding, we established a functional relationship between PD-related aberrant SNCA expression and mitochondrial dysfunction. In addition, we provided preclinical evidence that IR impairs mitochondrial function by suppressing PGC-1α expression, which elicits enhanced oxidative stress, increased PLK2 activity, and upregulated expression of phosphorylated and proteinase K-resistant SNCA, which is pathognomonic of PD. Conversely, the inhibition of PLK2 enzymatic activity or overexpression of PGC-1α reversed the IR-induced PD-eliciting/promoting biochemical cascade. A principal hallmark of PD is the deposition of Lewy bodies in DA neurons. SNCA is the main component of Lewy bodies [39]. In physiology, SNCA modulates intracellular vesicle formation, and its degradation is dependent on the ubiquitin–proteasome system and autophagy–lysosomal pathway [39]. However, hyperphosphorylation at specific amino acid residues facilitates abnormal folding, which interferes with degradation and results in oligomerization [40]. Several kinases are implicated in the hyperphosphorylation of SNCA, including PLK2 which phosphorylates SNCA at its Ser129 residue [41]. The present study demonstrated that the acquisition of an IR phenotype is associated with increased expression of total, Ser129-phosphorylated, and proteinase K-resistant SNCA, as well as PLK2/SNCA complex formation. This is in line with the enhanced PLK2 enzymatic activity observed upon induction of IR, and the converse repression of IR-enhanced SNCA expression sequel to the pharmacological inhibition of PLK2.

The enzymatic activity of PLK2 is regulated by several factors, including intracellular ROS [40]. Elevated intracellular ROS, otherwise known as oxidative stress, enhance the expression of PLK2 at mRNA and protein levels [42]. The mitochondria represent a principal source of intracellular ROS, and mitochondrial dysfunction, encapsulated by respiratory chain impairment, reduced biogenesis, and impaired mitophagy, is a vital component of PD pathogenesis [43]. We demonstrated that the ectopic (re)expression of PGC-1α impedes IR-induced mitochondrial dysfunction, represses IR-enhanced PLK2 activity, and downregulates erstwhile high SNCA expression. This is consistent with documented association of diabetes and IR with downregulated expression and/or activity of PGC-1α [44]. Contextually, PGC-1α is the master regulator of mitochondrial biogenesis and is dysregulated in PD [45]. The polymorphism of PGC-1α is a suggested risk factor of PD [46] and the expression of PGC-1α-dependent downstream targets has been shown to be lower in people with PD [47]. In addition, deregulated parkin (PRKN) expression and/or activity is pathognomonic to PD, however, parkin-interacting substrate (PARIS/ZNF746), a Krüppel-associated box-zinc finger protein (KRAB-ZFP) that accumulates in models of parkin (PRKN) inactivation and in the brain of patients with PD, has been shown to repress the expression of PGC-1α and its substrate, the nuclear respiratory factor (NRF)-1, by targeting insulin response sequences in the promoter regions of PGC-1α [48]. This in part corroborates our findings that IR elicits reduced PGC-1α protein level, increased ROS production, and enhanced depolarization of the mitochondrial membrane potential (ΔΨm), a triad that is characteristic of PD. Of translational relevance, we demonstrated that forced/ectopic (re)expression of PGC-1α confers resistance to IR-enhanced ROS and SNCA expression, even in the presence of augmented PLK2 activity, thus suggesting the therapeutic exploitability of altered PGC-1α expression as an efficacious neuroprotective strategy in patients harboring IR with high risk of PD.

Adding to the numerous roles ascribed to SNCA, including in the pathogenesis of PD, there are accruing suggestions of its role in the regulation of neuronal mitochondrial homeostasis [49,50,51]. Consistent with the findings of the present study, we posit a novel link between mitochondrial dysfunction and aberration in SNCA expression and/or activity. Inconsistent with contemporary knowledge on causality which stems from aberrant SNCA expression to mitochondrial dysfunction, we observed that in the context of IR, impaired mitochondrial function associated with an increase in ROS production helped stabilize PLK2, which in turn catalyzed the phosphorylation of SNCA at its Ser129 residue, consequently activating and enhancing the mitochondrial and nuclear accumulation of SNCA [52,53]. This is suggestive of an IR-associated mitochondrial dysfunction-PD vicious cycle.

As with most studies of this nature, the present proof-of-concept study has some limitations. The present study did not address neuroinflammation, which is another important pathological feature of PD [54,55]. Secondly, the SNCA expression may be affected by ubiquitin proteasomal activity. The present study did not specifically investigate whether IR impairs this activity. Thirdly, the small sample size of this proof-of-concept study invariably decreases its statistical power and may pose a challenge for generalizability.

In conclusion, our findings demonstrate that IR-associated diabetes promotes the development and progression of PD through PLK2-mediated mitochondrial dysfunction, upregulated ROS production, and enhanced SNCA signaling; thus, suggesting the therapeutic targetability of PLK2 and/or SNCA as potentially novel disease-modifying strategies in patients with PD.

## Figures and Tables

**Figure 1 cells-09-00740-f001:**
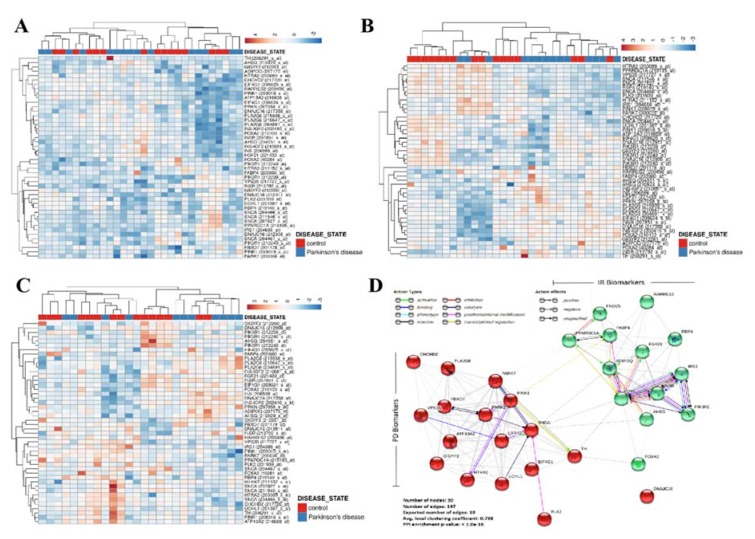
Patients’ genomic landscape indicates a nosological association between diabetes and Parkinson’s disease (PD), in silico. PD and insulin resistance (IR)-related transcript expression heatmaps from transcriptional analyses of (**A**) putamen (Homo sapiens, A-AFFY-33, AFFY_HG_U133A, E-GEOD-20291, 30 samples, 22,283 genes), (**B**) prefrontal area 9 (Homo sapiens, A-AFFY-33, AFFY_HG_U133A, E-GEOD-20168, 30 samples, 22,283 genes), and (**C**) whole substantia nigra (Homo sapiens, A-AFFY-33, AFFY_HG_U133A, E-GEOD-20292, 26 samples, 22,283 genes) in PD. Rows are centered; unit variance scaling is applied to rows. Both rows and columns are clustered using correlation distance and average linkage. (**D**) Visualization of the functional protein–protein interaction of PD and IR proteins as depicted by Search Tool for the Retrieval of Interacting Genes/Proteins database (STRINGdb).

**Figure 2 cells-09-00740-f002:**
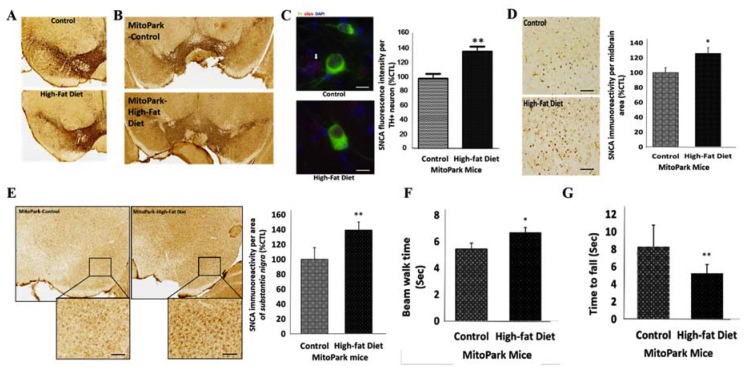
Diabetes contributes to the progression of PD by amplifying the loss of tyrosine hydroxylase (TH)^+^ dopaminergic neurons and enhancing α-synuclein (SNCA) expression, in vivo. (**A**) Representative IHC images of TH immunoreactivity in the midbrain substantia nigra of high-fat diet-induced diabetes C57BL/6 mice compared to the control mice. (**B**) Representative IHC images of SNCA immunostaining in the substantia nigra of diabetic MitoPark mice, compared to their control non-diabetic counterparts. (**C**) Representative immunofluorescence images of α-syn/SNCA (red) and TH (green) expression in the midbrain of control and diabetic MitoPark mice, captured by deconvolution microscopy. 4′,6′-diamidino-2-phenylindole (DAPI) serve as nuclear staining (left). Graphical representation of the SNCA fluorescence intensity in TH-positive dopaminergic (DA) neurons from diabetic MitoPark mice (control: 100 ± 6.6%, diabetic: 137.7 ± 8.2%, *n*_TH + neurons_ = 24, *p* < 0.01) (right). (**D**) Representative images (left) and graphical representation (right) of SNCA immunoreactivity in the midbrain tectum of control and high-fat diet-induced diabetes MitoPark mice (control: 100 ± 8.7%, diabetic: 126 ± 7.6%, *n* = 3). (**E**) Representative IHC images (left) and graphical representation (right) of SNCA expression in the substantia nigra of control and high-fat diet-induced diabetes MitoPark mice (control: 100 ± 15.7%, diabetic: 139.2 ± 10.9%, *n* = 3). *, *p* < 0.05, **, *p* < 0.01; control, non-diabetic mice. (**F**) Diabetic MitoPark mice took a longer time to walk the beam compared with control MitoPark mice (control: 5.43 ± 0.46, diabetic: 6.67 ± 0.39 s, *n* = 3). (**G**) Diabetic MitoPark mice stayed a shorter length of time on the rotarod tests (control: 8.3 ± 2.5, diabetic: 5.3 ± 1 s, *n* = 3).

**Figure 3 cells-09-00740-f003:**
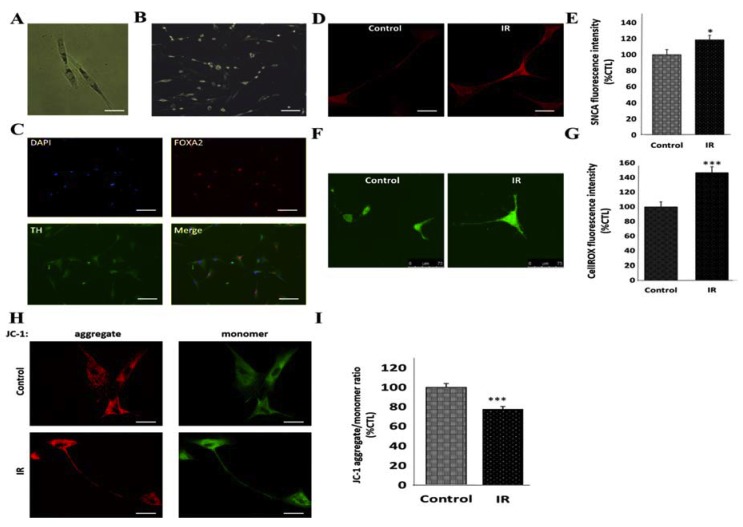
Insulin resistance is associated with increased expression of SNCA and mitochondrial dysfunction in human differentiated dopaminergic neurons, ex vivo. Photo images of (**A**) the fusiform fibroblast-like adipocyte-derived stem cells (ADSCs) obtained from the adipose tissue of non-diabetic, non-PD subjects, and (**B**) neuron-like morphology with many neurite outgrowths, on day 21 of ADSC neuronal differentiation. (**C**) Immunofluorescence images of d-hDA neurons characterized by TH (green) and FOXA2 (red) immunopositivity and outgrowth of neurites. (**D**) Representative images and (**E**) histograms of SNCA immunofluorescence intensity in d-hDA neurons from patients with or without IR (control: 100 ± 6.3%, IR: 118.4 ± 5.6%, *p* < 0.05, *n* = 12). (**F**) Representative images and (**G**) histograms of CellROX live staining of d-hDA neurons from patients with IR compared to non-IR control, captured by confocal microscopy (control: 100 ± 7%, IR: 146.6 ± 8.1%, *p* < 0.001, *n* = 10). (**H**) Representative images and (**I**) graph of JC-1 (5,5′,6,6′-tetrachloro-1,1′,3,3′-tetraethylbenzimidazol-carbocyanine iodide) aggregate/monomer live staining of d-hDA neurons from patients with IR compared to non-IR control. Reduced JC-1 aggregate (red)/monomer (green) ratio indicates the ΔΨm depolarization in the d-hDA neurons (control: 100 ± 4.1%, IR: 77.8 ± 3%, *p* < 0.001, *n* = 12). *, *p* < 0.05, ***, *p* < 0.001; d-hDA, differentiated human dopaminergic neuron; IR, insulin resistance; ΔΨm, mitochondrial membrane potential.

**Figure 4 cells-09-00740-f004:**
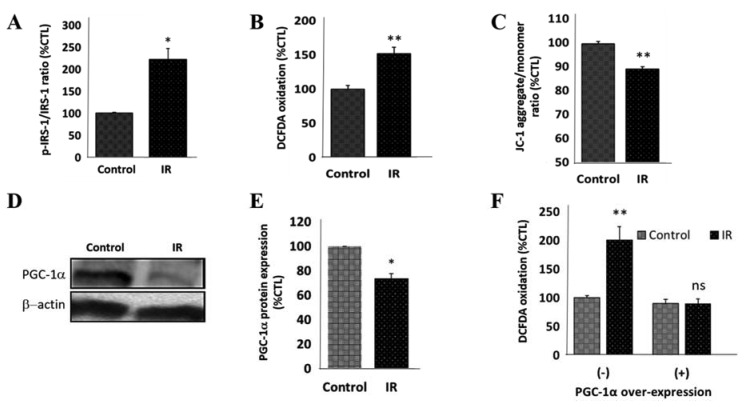
Peroxisome proliferator-activated receptor gamma coactivator 1-alpha impedes IR-induced mitochondrial dysfunction. Graphs of the differential (**A**) p-IRS-1/IRS-1 ratio (control: 100 ± 1.7%, IR: 223 ± 25%, *p* < 0.05, *n* = 4), (**B**) DCFDA (2′,7′-dichlorodihydrofluorescein diacetate) oxidation (control: 100 ± 5.3%, IR: 152.1 ± 8.9%, *p* < 0.01, *n* = 4), and (**C**) JC-1 aggregate/monomer ratio (control: 100 ± 1.5%, IR: 88.9 ± 0.8%, *p* < 0.01, *n* = 4), in IR SH-SY5Y cells, compared to the non-IR control cells. (**D**) Representative Western blot images and (**E**) densitometry-based histograms showing differential proliferator-activated receptor gamma coactivator 1-alpha (PGC-1α) protein expression in IR and non-IR control SH-SY5Y cells (control: 100 ± 0%, IR: 73.4 ± 4.1%, *p* < 0.05, *n* = 4). β-actin is the loading control. (**F**) Graph showing the effect of PGC-1α o/e on the differential DCFDA oxidation level in IR and non-IR control SH-SY5Y cells. ns, *p* > 0.05, *, *p* < 0.05, **, *p* < 0.01; IRS, insulin receptor substrate; o/e, overexpression.

**Figure 5 cells-09-00740-f005:**
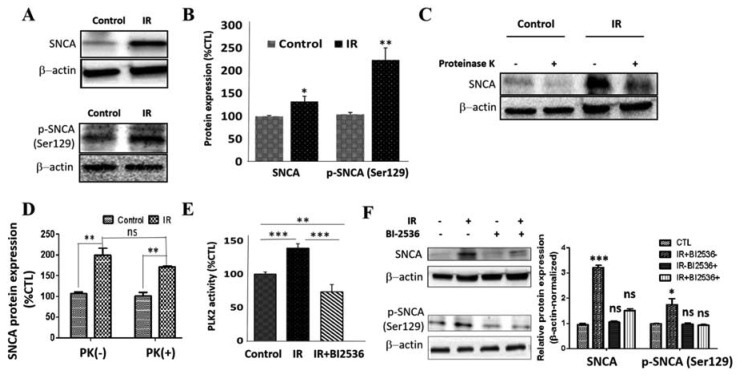
IR elicits increased total, Ser129-phosphorylated, and proteinase K-resistant SNCA through a polo-like kinase 2-dependent mechanism. (**A**) Representative Western blot images and (**B**) densitometry-based graph showing the effect of IR on the expression of total (upper) and Ser129-phosphorylated (lower) SNCA proteins in SH-SY5Y cells (SNCA, control: 100 ± 1.8%, IR: 132.1 ± 11.3%, *p* < 0.05, *n* = 6; p-SNCA (Ser129): control: 100 ± 3.8%, IR: 214.3 ± 26%, *p* < 0.01, *n* = 6). (**C**) Representative Western blot images and (**D**) densitometry-based histograms showing the effect of proteinase K on the expression of SNCA protein in IR and non-IR control SH-SY5Y cells. (**E**) Graphical representation of the effect of 0.0625 nM BI-2536 on polo-like kinase-2 (PLK2) activity in IR SH-SY5Y cells, compared to non-IR control cells (control: 100 ± 3.3%, IR: 138.9 ± 6.4%, IR + BI2536: 73.4 ± 10.9%, *n* = 6). (**F**) Representative Western blot images and histograms showing the effect of BI-2536 on the expression of SNCA (upper) and p-SNCA (Ser129) (lower) proteins in SH-SY5Y cells. β-actin was used as a loading control; ns, *p* > 0.05, *, *p* < 0.05, **, *p* < 0.01, ***, *p* < 0.001; BI-2536, selective PLK2 inhibitor.

**Figure 6 cells-09-00740-f006:**
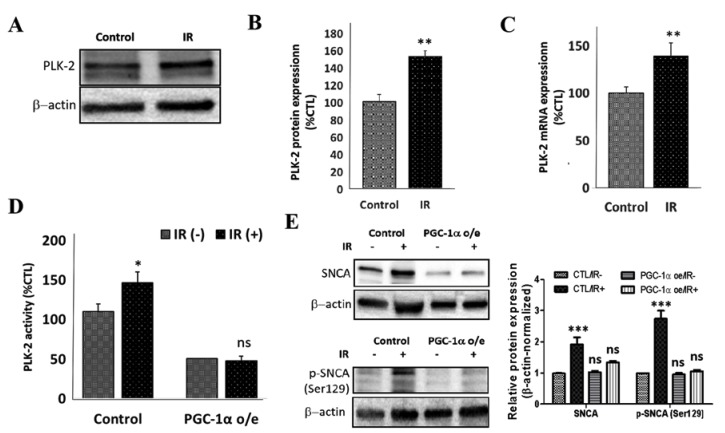
PGC-1α overexpression represses IR-enhanced PLK2 activity and subsequent increase in SNCA. (**A**) Representative Western blot images and (**B**) densitometry-based graph showing the effect of IR on the expression of PLK2 protein in SH-SY5Y cells (control: 100.5 ± 8.4%, IR: 152.9 ± 6.9%, *p* < 0.01, *n* = 5). (**C**) Graphical representation of the effect of IR on the expression of PLK2 mRNA level in SH-SY5Y cells (control: 100.4 ± 6.3%, IR: 139.3 ± 13.9%, *n* = 4). (**D**) Histograms showing the effect of PGC-1α o/e on PLK2 activity in IR or non-IR SH-SY5Y cells (control cells, IR (−): 100 ± 9.5%, IR (+):146 ± 14%, *p* < 0.05; PGC-1α o/e cells: IR (−): 50.5 ± 0.5%, IR (+): 48 ± 6%, *p* > 0.05, *n* = 4). (**E**) Representative Western blot images and histograms showing the effect of PGC-1α o/e on the expression of SNCA (upper) and p-SNCA (Ser129) (lower) proteins in IR SH-SY5Y cells compared to their non-IR control counterparts. β-actin was used as loading control; o/e, overexpression; ns, *p* > 0.05, *, *p* < 0.05, **, *p* < 0.01, ***, *p* < 0.001.

**Figure 7 cells-09-00740-f007:**
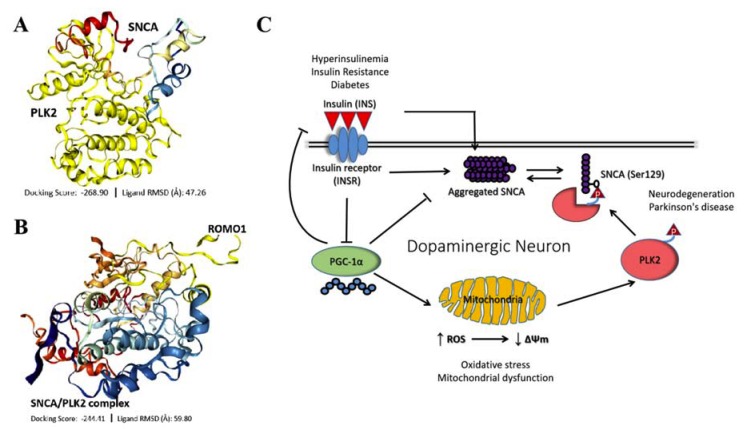
IR-driven SNCA-PLK2-reactive oxygen species (ROS) signaling is implicated in PD development and progression. Molecular docking showing direct interaction between (**A**) SNCA and PLK2 with shape complementarity/docking score of –268.90, and ligand root-mean-square deviation (RMSD) of 47.26Å and (**B**) SNCA-PLK2 complex and reactive oxygen species modulator 1 (ROMO1) with a docking score of –244.41, and ligand RMSD of 59.80Å. (**C**) Schematic abstract illustrating an association between IR and PD, with IR promoting PD through aberrant expression of SNCA, enhanced mitochondrial oxidative stress, mitochondrial dysfunction, and deregulation of the PLK2 signaling.

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
