# Peer review of "Insulin Resistance Promotes Parkinson’s Disease through Aberrant Expression of α-Synuclein, Mitochondrial Dysfunction, and Deregulation of the Polo-Like Kinase 2 Signaling"

_cells, 2020, doi:10.3390/cells9030740_

Round 1
Reviewer 1 Report
This is a novel and timely study exploring the links between neurodegeneration and insulin resistance and the authors should be commended on their approach. However there were a few points that were not clear that would benefit from further clarification before the article could be accepted for publication:
- the DA differentiation - what percentage of neurons were dopaminergic? Were they electrically functional?
- How did the authors induce insulin resistance in the h-DA neurons? the protocol was not clear. How did the authors confirm that this indeed induce insulin resistance (defined as the reduced responsiveness of tissues to the action of insulin) in h-DA neuons? Did the neurons respond differently to hyperglycaemic states or purely hyperinsulinaemic states?
- Which IRS-1 phsoporylated residue was measures against IRS-1
Reviewer 2 Report
The study by Hong et al seeks to demonstrate that insulin resistance (IR) may contribute to Parkinson’s disease (PD) through an interwoven relationship between mitochondrial dysfunction, alpha synuclein expression, and PLK2 activity. Using MitoPark mice as a model of PD, the authors first suggest that a high fat diet increases alpha synuclein, decreases dopaminergic neuron populations, and increases motor deficits. The authors then use in vitro systems to show that IR increases oxidative stress, alpha synuclein expression, and PLK2 expression and activity, and furthermore these can be blocked through overexpression of PGC-1a. Given the already well known link between IR and protein aggregation, and IR and mitochondrial dysfunction, it is unclear what precisely the current study is adding to the wider knowledge surrounding IR and PD. The authors themselves state in the first paragraph of the discussion “Chronic hyperinsulinemia induces IRS-1 hyperactivity in neurons and eventually leads to IR with its associated pathogenic effects, including mitochondrial dysfunction, oxidative stress, abnormal glucose metabolism, inflammation, protein aggregation, and defective neurogenesis.” Are the authors attempting to be the first to show this effect may be mediated by PLK2? Or demonstrate the effect specifically in dopamine neurons? i.e what is the key finding from this study? The answer to this question should be made clearer in both the introduction and discussion sections
Some additional major concerns that also need to be addressed are outlined below.
Methods
There is currently a critical amount of information missing from the methods section that would make it extremely difficult for researchers to replicate experimental paradigms.
- How IR was induced in vitro is missing
- No information regarding genomic analysis. What tools/packages were used? How was String used?
- Methods for Figure 7 missing.
- Section 2.6: “After cells were collected by centrifugation…” Which cells? Did the lysis buffer contain protease inhibitors?
- Section 2.10: How was the tissue processed? Was is perfused, cryoprotected? How thick were the sections that were cut?
- Section 2.11: Were the cells fixed? How was densitometry performed? Stating it was done through ImageJ is not sufficient
- Section 2.12: This section missing a lot of information that is deemed essential by MIQE guidelines for publicatino
Results
- The authors need to clarify their sample size descriptions throughout the manuscript, particularly for the in vivo experiments. The methods state that 6 x C57BL6 and 6 MitoPark mice were purchased, and then divided into 6 diabetic and 6 non-diabetic groups. This would mean there was n=3/group, however Figure 2 states n=24 for panels C and D and n=4 for panel E.
- N=3/ group is extremely insufficient for behavioural testing. Further, it is surprising that p<0.01 was achieved given the large error bars for the control group in Fig 2G and the small sample size.
- The authors need to be consistent with how they report p values, with some written as p = XYZ while other described as p<0.05. Also there are a number of instances where the p value given does not match graph (eg line 331).
- Can the authors provide evidence that 8 weeks of HFD is sufficient to produce IR or diabetes? This is a particularly short period of time compared to many HFD protocols, and without results provided from glucose tolerance test, insulin tolerance test, body weight changes etc, there is no evidence that IR/diabetes has been induced in the in vivo experiments.
- Fig 2B states images are from MitoPark mice, yet text within results section describes it as showing increase in number of cells expressing alpha synuclein in C57Bl6 mice. Further, Fig 2E is also representing staining of alpha synuclein, yet this stain looks vastly different to that in 2B.
- Graph in Fig 2D suggests number of TH cells decrease in HFD group, yet images would suggest opposite. How was this data obtained? Was only 1 slice analysed or multiple throughout the SN? Did the analyses only encompass the SNpc?
- Fig 3 legend: did the cells originate from patients with IR vs healthy controls, or was IR induced in vitro?
- No associated graphs showing quantification of Fig 5F and 6E.
- There are some instances where the gel images for western blots are not representing the data are presented. For example, Fig 5C shows alpha synuclein decrease in the IR + proteinase K treated group, but Fig 5D suggests it should increase. In Fig 5F there is not a large increase in p-SNCA in the IR group which is surprising given the 114% increase in p-SNCA in Fig 5B. How consistent are these changes?
- Figure 7: with the relevant methods section missing, where is the evidence that these formations are occurring directly in relation to IR vs normal conditions?
Discussion
Some general thoughts is that the discussion does not really touch on the literature that has already looked at insulin resistance and a-syn/mitochondria/PGC-1alpha. What is this study trying to be the first to show and how is it different to previous studies? Careful attention should be made so as to not overstate the results of the study, particularly inferring any causal relationships. In a number of instances, statements are made (lines 482-482, lines 492-494, lines 524-525 relating to mitochondrial and nuclear accumulation) that cannot be supported as the data is currently missing from the study.
Also missing is any discussion of the results presented in Figures 1 and 2.
Minor Point
- Ref 5 and 6 in the reference list are the same reference but has been split over 2 numbers. Same for Ref 24 and 25. This has meant that the reference numbers throughout the manuscript are off slightly.
Reviewer 3 Report
This manuscript investigated potential mechanisms for how impaired insulin signaling may promote PD, using cell cultures and mouse models of PD and diabetes. The authors acknowledge in the introduction that the relation in humans between diabetes and PD is tenuous but are fairly clear throughout that they are studying the mechanistic basis of a possible link rather than claiming evidence of cause and effect between diabetes and PD. However, the small N and unclear grouping of the mouse models causes some concern as to how conclusive the animal results can be and is not noted as a limitation.
There may still be some resistance to calling Alzheimer's Type 3 diabetes, as there was no evidence of brain hyperglycemia until a few recent Magnetic Resonance Spectroscopy and Postmortem studies showed these features. Even so, using "type 3 diabetes" elicits controversy and may best be left out unless the authors have strong feelings about using them.
6 and 6 is very low N. I cannot tell from 2.1 how these groups were further split into HFD or control diet. I can see these as being supplemental but the low N does not seem to be noted as a limitation. Any explanation of how they were split (evenly?) into groups is a necessity.
Round 2
Reviewer 2 Report
This reviewer thanks the authors for outlining responses to the points raised in the first review of this manuscript. The majority of these points have now been adequately addressed and improve the quality of the manuscript. There are some issues outlined below that still require attention.
- Regarding the confusion from the authors about what is requested for the bioinformatics analysis, it is the understanding of this reviewer that the authors have accessed data from previous studies and then performed their own analysis on this data. Is this correct? If so, then it is assumed the data was analysed in a different way in this current study to how it was analysed in the already published studies. The authors therefore should be stating in a separate section within the Methods (and not buried within the Results) what their bioinformatics approaches were, such as what datasets were accessed, what software or online tools were used to analyse the data, any appropriate parameters utilised etc to allow for any other researcher to replicate their findings. This also applies to the STRING analysis and the molecular docking results from the manuscript
- Section 2.6 states ADSC-derived cells were collected and processed for western blot, but there does not appear to be any western blot data for these cells, only for the SH-SY5Y cells.
- The authors have not addressed the differences in p value reporting throughout the manuscript. There are still numerous occasions in which an exact p value is given (for eg p=0.04 or p=0.01) rather than within a range (eg p<0.05, p<0.01). Further, there are still multiple occasions in which the p-values written in the text do not match the significance values presented in the graphs. Examples of both include lines 370, 372, 384, 400, 413, 422, 424, 430, 434, 440, 456, 466, 467, 480, 483
- Can the authors please provide further clarification regarding Figure 2C? The graph has changed since the first submission, with the high fat MitoPark mice having ~38% increase in SNCA fluorescence compared to controls while in the previous submission this was stated as ~26% increase. Have the authors re-analysed this particular data set? If so why was the high fat diet group the only one re-analysed (the control 100% +/- 6.6% has not changed). However we appreciate that it has now been made clear that n=24 refers to the number of TH+ neurons analysed
- Can the authors please provide clarification regarding Figures 2D and 2E? In the previous submission Figure 2D was stated as showing TH immunoreactivity with the HFD mice showing a decrease in the graph. Figure 2D now describes SNCA immunoreactivity with the HFD mice showing an increase. How then is the data presented in Figure 2D different to the data presented in Figure 2E (which also shows SNCA immunoreactivity)? The authors state in their response letter that the midbrain substantia nigra was the area analysed, so again it is unclear how the 2 panels are showing anything different?
- With no analysis of TH positive cell numbers or analysis of levels of TH staining, the authors should remove any reference to evidence of increased loss of TH+ cells throughout the manuscript and also change the title of Figure 2.
- Legend for Figure 2 references BALB/c mice instead of C57BL/6
- We thank the authors for including new a gel image for p-SNCA in Fig 5F that is more representative of the data. However it would appear that the b-actin image for Fig 5F has not also been changed. Can the authors provide the correct b-actin gel image so the bands are from the same samples?
- We previously raised that gel images for Fig 5C did not represent the data presented. This does not appear to be addressed
- in Figure 5D there is no reduction in SNCA protein expression in the control + proteinase K group. As such, how can the authors be sure the increased SNCA expression in the IR+Prot K is because the alpha synuclein is resistant to PK and not just due to the increase SNCA expression induced by IR, a finding that the authors show multiple times throughout the manuscript. If this can not be resolved the authors may need to remove references that IR upregulates expression of proteinase K-resistant SNCA from the manuscript
